# Functional duality in group criticality via ambiguous interactions

**Takayuki Niizato**[1]*, **Hisashi Murakami**[2], **Takuya Musha**[1]

**1** Faculty of Engineering, Information and Systems, University of Tsukuba, Tsukuba, Ibaraki, Japan, **2** Faculty of Information and Human Science, Kyoto Institute of Technology, Sakyo-ku, Kyoto city, Kyoto, Japan

* niizato@iit.tsukuba.ac.jp

**Data Availability Statement:** All code and data are available in the Matlab code (for PID, https://github.com/robince/partial-info-decomp) and in Supporting information.

**Funding:** Support for data collection was provided by the Grants-in-Aid for Scientific Research from

## Abstract

Critical phenomena are wildly observed in living systems. If the system is at criticality, it can quickly transfer information and achieve optimal response to external stimuli. Especially, animal collective behavior has numerous critical properties, which are related to other research regions, such as the brain system. Although the critical phenomena influencing collective behavior have been extensively studied, two important aspects require clarification. First, these critical phenomena never occur on a single scale but are instead nested from the micro- to macro-levels (e.g., from a Lévy walk to scale-free correlation). Second, the functional role of group criticality is unclear. To elucidate these aspects, the ambiguous interaction model is constructed in this study; this model has a common framework and is a natural extension of previous representative models (such as the Boids and Vicsek models). We demonstrate that our model can explain the nested criticality of collective behavior across several scales (considering scale-free correlation, super diffusion, Lévy walks, and $1/f$ fluctuation for relative velocities). Our model can also explain the relationship between scale-free correlation and group turns. To examine this relation, we propose a new method, applying partial information decomposition (PID) to two scale-free induced subgroups. Using PID, we construct information flows between two scale-free induced subgroups and find that coupling of the group morphology (i.e., the velocity distributions) and its fluctuation power (i.e., the fluctuation distributions) likely enable rapid group turning. Thus, the flock morphology may help its internal fluctuation convert to dynamic behavior. Our result sheds new light on the role of group morphology, which is relatively unheeded, retaining the importance of fluctuation dynamics in group criticality.

## Author summary

To investigate the critical phenomena influencing collective behavior, we propose the ambiguous interaction model as a natural extension of the Boids model. Our proposed model exhibits various critical properties with respect to real-world collective behavior depending on the parameter settings (scale-free correlation, Lévy-walk behavior, and $1/f$ fluctuations in the center-of-mass frame). The results show that individual and group criticalities originate from the single algorithm employed by our model.

the Japan Society of the Promotion of Science (21H05302 to T.N.). The funders had no role in study design, data collection and analysis, decision to publish, or preparation of the manuscript.

Furthermore, we determine the functional duality for different input types (velocity and fluctuation) using a scale-free induced correlated domain inside a flock. The information flow between sub-domains within the flock is found to be bidirectional rather than unidirectional. This means that, contrary to appearances, a flock does not have a leader-follower information structure. Moreover, our analysis suggests a strong relationship between group morphology (i.e., velocity distributions) and its fluctuation power (i.e., fluctuation distributions) for rapid group turning. Our result also sheds new light on the role of group morphology, which has not been thoroughly investigated, retaining the importance of fluctuation dynamics in group criticality.

## Introduction

The critical phenomena of collective animal behavior, which are widely observed [1–4], elucidate the criticality of living systems [5–8]. However, the study of these phenomena is hindered by two core, interrelated problems: (1) the critical phenomena are nested across several scales, and (2) the functional roles of group criticalities are not clearly understood from an adaptive perspective. The first problem reveals the statistical properties required for systems analyzed in research studies. The second problem indicates that if the role of criticality is not well understood, it is possible that it is not treated with sufficient importance.

Criticality in collective behavior occurs on at least two levels: macro and micro. The macro-scale criticality is the criticality of the system as a whole, as represented by the scale-free correlation of the system [8–12]. Cavagna et al. [9] found that a flock (of starlings, for example) has size-independent correlation domains for its direction and speed, and that high correlation raises the system susceptibility [13–15]. Thus, with scale-free correlation, the members of the flock can quickly share information and achieve optimal responses to external stimuli as a group. Overall, macro-scale criticality maximizes the benefits of group membership in various ways [6–8]. In contrast, micro-scale criticality is the one that occurs for an element or individual as represented by a Lévy walk [16–23], which is an optimal search strategy (i.e., the optimal balance between exploration and exploitation) for a given space. Some researchers have also suggested that Lévy walks contribute to smooth communication between individuals in flocks [24, 25]. Notably, each fish in a group tries to search for and communicate with the new neighbors within the group. The micro-scale criticality may also therefore be related to group benefits.

Although these critical properties have been separately reported, several systems seem to coexist (e.g., in bacteria [10, 18], fish schools[12, 24], and proteins[11, 22]). At first glance, group and individual benefits appear to be in conflict each other, because there is generally a trade-off between the two [26, 27]; however, the nested critical system seems to address these disagreements. The concept of "nested criticality" considered in this paper is derived from these observations.

Next, we consider the second problem noted above, i.e., the lack of clarity regarding the relationship between criticalities and the observed collective behavior. Group turning is one of the most debated topics of collective behavior. Although some researchers have suggested that criticalities and group turning are related [9, 28], their suggestions seem to be more suitable to options other than criticality. For instance, Attanasi et al.[29, 30] believe that the quantum effect causes rapid group turning and have also suggested that individual fluctuations may trigger this behavior [31]. These solutions are attractive in themselves; however, the relationship between collective behavior and critical phenomena remains obscure. Unfortunately, the

currently existing technologies do not provide sufficient data (only time intervals of a few seconds for analysis [31, 32]) to describe the relationship between group turning and critical phenomena.

In a previous study, we developed a collective behavioral model based on ambiguous interactions as one possible solution [33]. We applied this model to several flocking models and showed that the interaction between the alignment and attraction can be regarded as a single interaction with time-scale differences. Alignment pertains to the infinitely long predicted neighbor positions of a given agent, whereas attraction relates to the interactions of that agent with the current neighbor positions. We showed that the medium region of this time scale could play a vital role in group criticality. However, our result was restricted to a two-dimensional system, and we did not investigate the functional role of criticality. Additionally, as positional relations in three dimensions are more complex than those in two dimensions, it was unclear whether our model could yield meaningful results for the three-dimensional case. Especially, the noise definition of our model must be reconsidered for further extension to general situations.

In the present study, we propose a model that can simultaneously explain nested criticality in collective behavior and its functional abilities (i.e., group turning); our model is a natural extension of our previous model to three dimensions. The self-tuned noise through ambiguous interactions make it possible to bridge the gap between micro- and macro-scale criticality. Moreover, our newly developed model elucidates the two features discussed above (i.e., the nested criticality and its functional roles) via an information theoretic analysis. The remainder of this paper is structured as follows. First, we present our model algorithm and explain why our model is a natural extension of the previous representative models. Second, we examine the criticality of group behavior. For various kinds of criticality (super diffusion, scale-free correlation, Lévy walks, and $1/f$ fluctuation), we confirm that nested criticality holds without strict parameter tunings. This suggests that the critical phenomena are universal and can be observed in the broader range of parameter regions rather than only in a restricted region as conventionally thought. Third, we divide a flock into two scale-free induced subgroups and examine the information transfer between them, evaluating the influence of this coarse-grained information (the average vectors of the fluctuation and velocity vectors) on the future behavior of the group. Our analysis reveals that these coarse-gained average vectors significantly impact the rapid group turning behavior. Finally, we present a new method of applying partial information decomposition (PID) [34, 35] to the group behavior and find that combining group morphology (i.e., the velocity distribution) with internal fluctuations can likely enable rapid group turning. Our findings imply that flock morphology may aid in the conversion of internal fluctuations into dynamic behavior.

## Materials and methods

First, we briefly review the main concepts used in our model and highlight the ones that are based on a natural extension of basic flocking models, such as the self-propelled particle model and the Boids model [36–38].

Most flocking models involve two interactions: alignment and attraction [39–46], which have been experimentally confirmed [2, 47–50]. With regard to alignment, each agent changes its direction to align with its chosen neighbors (e.g. within the certain radius[37] or the fixed topological distance[2]), whereas in the case of attraction, each agent moves toward the current positions of its neighbors. Many researchers have argued that a balance between alignment and attraction is key to the various flocking formations observed in nature [38, 42, 47], and recently, the reducibility of one of these interactions to the other has been highlighted [46].

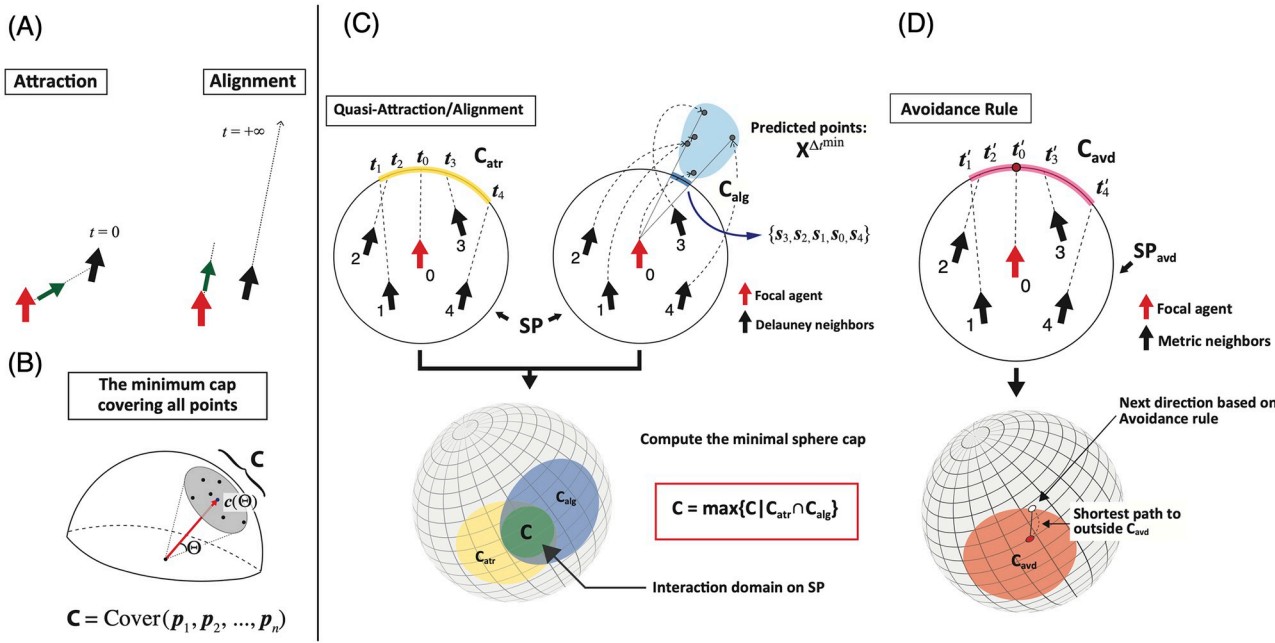

**Fig 1. Brief sketch of the quasi-attraction/alignment algorithm.** (A) Reinterpretations of attraction ($t = 0$) and alignment ($t = \infty$). The red and black arrows indicate the focal agent and its neighbor, respectively, while the green arrow indicates the next direction. (B) A sketch of the cover function, which returns the minimum cap C on the interaction sphere SP, which covers all points $p_i$ (for the mathematical definition, see the Section 2 in S1 Appendix). (C) Left: The quasi-attraction projected on the plane. Each direction is extended to SP, where $C_{atr}$ (yellow) is the minimal sphere cap covering all points $t_j$. Right: The quasi-alignment projected on the plane, where $C_{alg}$ (blue) is the minimal sphere cap covering all points $s_j$. Lower: C (green) is the maximal sphere cap inside $C_{atr} \cap C_{alg}$. The focal agent selects its next direction randomly based on C. (D) A brief sketch of the avoidance algorithm. Upper: Each direction is extended to the repulsion area $SP_{avd} = \{r | |r| = R\}$, where $C_{avd}$ is the minimal sphere cap that covers all points on $SP_{avd}$. (Lower) The focal agent determines its next direction uniquely as the shortest-distance position to the outside of $C_{avd}$.

Based on these findings, our claim is clear: both the interactions stem from a common interaction.

As another definition of alignment, the process by which one agent aligns in parallel with its neighbors can be described as follows: each agent tries to adjust its direction to the infinite future positions of its neighbors, assuming that the neighbors will continue to follow their current trajectories. Similarly, attraction can be described as the agent adjusting its direction according to the current positions of its neighbors (Fig 1A). Thus, the difference between alignment and attraction lies in the prediction time scale (for more detail, see [33]). Thus, the difference between these two interactions is not with regard to the type but the degree, and we observe the same type of interaction across different time scales. This analysis suggests new possibilities. Between the two extremes (i.e., current and infinite futures), our flocking model provides various grades of interactions.

## Quasi-Attraction and Quasi-Alignment

Based on the aforementioned considerations, we define a new flocking model that naturally extends the previous models. In other words, because we have confirmed that two extremes (attraction and alignment) for the flocking model correspond to different time scales, our aim is to find smooth connections between them.

Let us define the "interaction sphere" SP for the connection (Fig 1B). First, we draw a Delaunay diagram from the positional relationships of the given agents [51]. The neighbors are connected to all the Delaunay triangles. Second, we compute the distance of each neighbor

$j$ from the agent of interest $i$, i.e., $D_{ij}$, and take its maximum. This value is the radius of SP. Mathematically, the interaction sphere of agent $i$, $\mathsf{SP}_i = \{r \in \mathbb{R}^3 | |r| = R^i_{max}\}$, where $R^i_{max} = \max_{j \in \mathsf{N}_i}\{D_{ij}\}$ and $\mathsf{N}_i$ is the set of Delaunay neighbors of $i$. Thus, $\mathsf{SP}_i$, covers all neighbors directly linked with its Delaunay triangles.

Note that SP contains interaction domains from which the agent of interest determines its next direction. In our model, unlike most other models, we do not adopt the average values with external noise. Instead, we define a vaguer region on the interaction sphere; that is, a spherical cap C. The possible directions must be selected from within this vague region (Fig 1B). This sphere enables the agents to tune their randomness according to their situations.

Next, we define the quasi-attraction, which consists of relatively small time-scale predictions such as genuine attraction ($t = 0$). First, the focal agent $i$ determines $\mathsf{SP}_i$ from its Delaunay neighbors. The next step involves calculating the point at which the extended velocity vectors $v_j$ intersect with $\mathsf{SP}_i$; i.e., $t_j$ ($\forall j \in \mathsf{N}_i$: Fig 1B: left). Here, we define the minimal spherical cap $\mathsf{C}_{atr}$, which covers all these intersection points ($t_i, t_1, t_2, \ldots, t_{|\mathsf{N}_i|}$). This cap can be computed using the cover function (Fig 1B); that is, $\mathsf{C}_{atr,i} = \mathrm{Cover}(t_i, t_1, t_2, \ldots, t_{|\mathsf{N}_i|})$. We also note that the cap can be defined as having two factors, the solid angle $\Theta$ and the cap center $c(\Theta)$, as shown in Fig 1B.

This cap $\mathsf{C}_{atr,i}$ is the interaction domain induced by the quasi-attraction (Fig 1C: left). We use the prefix "quasi-" because these intersections indicate longer time scales than that obtained for pure attraction. Here, we consider the time scales in the context of the prediction from the inside sphere. Although the travel time is relatively short, it is not necessarily $t = 0$. The travel time depends on the neighbor relative positions and the current direction of the focal agent only. For instance, in Fig 1C: left, agent 3 takes shorter time to achieve SP than agent 1. If the agent is on the SP, the reaching time becomes minimum, that is, $t = 0$.

In contrast, quasi-alignment pertains to long time-scale predictions. Recently, some studies have suggested the importance of future predictions of flocking movement [43, 46, 52–55]. In our model, each agent predicts the behavior of its neighbors based on their past movements; that is, their turning rates [33].

The turning rate can be represented using the rotation matrix R, which is defined as follows. According to the Rodrigues' rotation formula [56], two factors determine the velocity rotation in three dimensions: the axis and the angle. The axis is in the direction that is unchanged by the vector rotation and the angle is the degree of rotation around that axis. If we have two vectors, we can construct R (Section 3 in S1 Appendix), and using R, we can then uniquely predict the future direction from the current direction. For example, if the current unit velocity vector is $\hat{v}^t_j$ (Fig A in S1 Appendix), the future direction is expected to be $\mathrm{R}^t_j\hat{v}^t_j$.

For the quasi-alignment, we consider several prediction time scales $\Delta t$. The current direction can be described in several ways depending on $\Delta t$, with $v^{t,\Delta t}_j = r^t_j - r^{t-\Delta t}_j$. The predicted future position $r^{t+\Delta t}_{\mathrm{pre},j} = r^t_j + (v_{\max}\Delta t)\mathrm{R}^{\Delta t}_j\hat{v}^{t,\Delta t}_j$, where $r^t_j$ is the current position. From this expression, if the agent behavior is predicted for long time scales (i.e., large $\Delta t$), $r^{t+\Delta t}_{\mathrm{pre},j}$ is a great distance from $r^t_j$. As the quasi-alignment yields predictions for long $\Delta t$, we compute all $r^{t+\Delta t}_{\mathrm{pre},j}$ extending beyond $\mathsf{SP}_i$; that is, $||r_i - r^{t+\Delta t}_{\mathrm{pre},j}|| > R^i_{max}$ ($\forall j \in \mathsf{N}_i \cup \{i\}$, including self-prediction).

Let $\Delta_{\min}$ represent the minimum time step that satisfies the above-mentioned condition. Then, we have a set $\mathsf{X}^{\Delta_{\min}} = \{r^{t+\Delta_{\min}}_{\mathrm{pre},j} | \forall j \in \mathsf{N}_i \cup \{i\}\}$ (Fig 1C: right). By referring to $\mathsf{X}^{\Delta_{\min}}$, each prediction point $s^t_j$ on $\mathsf{SP}_i$ can be determined (see Section 3 in S1 Appendix). Finally, we also define the interaction domain $\mathsf{C}_{alg,i}$ ($= \mathrm{Cover}(s_i, s_1, s_2, \ldots, s_{|\mathsf{N}_i|})$) from its distribution on $\mathsf{SP}_i$ (Fig 1C: right).

The maximal sphere cap on $\mathsf{SP}_i$, $\mathsf{C}_i$, is included in $\mathsf{C}_{atr,i} \cap \mathsf{C}_{alg,i}$ and gives the next direction of the focal agent (Fig 1C: lower):

$$\mathsf{C}_i = \max\{\mathsf{C}|\mathsf{C} \subset \mathsf{C}_{atr,i} \cap \mathsf{C}_{alg,i} \subset \mathsf{SP}_i\} \tag{1}$$

Let $c(\Theta_i)$ be a cap center and $\Theta_i$ be a solid angle of $\mathsf{C}_i$ along $c(\Theta_i)$. We set the condition that a random direction directs the point, obeying a Gaussian distribution with variance $\Theta_i$ along the mean axis of $c(\Theta_i)$. On the sphere, the Gaussian randomness is given by the von Mises–Fisher distribution [57–59]. This function determines a random output based on a given mean $c(\Theta_i)$ and variance $\Theta_i$ (note that random values can also be outside the cap $\mathsf{C}_i$). We define the random function $\mathrm{Random}_{VM}(\Theta_i, c(\Theta_i))$, which obeys the von Mises–Fisher distribution (see Section 4 in S1 Appendix for the precise definition). In other words, the randomness intensity depends completely on the neighbor behaviors (the degree of $\Theta_i$): if the neighbors have high alignment (i.e., small $\Theta_i$), the noise naturally degrades, but if they have low alignment (large $\Theta_i$), the noise increases. The noisy behavior of the focal agent is self-tuned according to its environment. Following this computation, each velocity is determined as follows:

$$v_i = v_{max}\cos(d\theta_i/2) \tag{2}$$

From this equation, it is clear that the deceleration depends on the degree of each turning direction ($d\theta$). This condition is based on experimental data [24]. We only use the fact that the individual turn decelerates the agent speed from this study.

Finally, we define avoidance. In this study, avoidance rarely occurs because the agent speed exceeds that of the avoidance domains occurring in real-world situations. The definition of avoidance is opposite to that of quasi-attraction. In other words, the avoidance determines the direction in which the agent must travel to avoid the sphere cap region created by quasi-attraction, i.e., $\mathsf{C}_{avd,i} (= \mathrm{Cover}(t'_i, t'_1, t'_2, \ldots, t'_n))$, where $n$ is the number of metric neighbors; Fig 1D: upper). The next direction given by the avoidance rule is the point of shortest distance to the edge of this sphere cap. Mathematically,

$$\arg \min_{r \in \partial \mathsf{C}_{avd,i}} \mathrm{dist}_{\mathsf{SP}_{avd}}(t'_i, r)$$

where $\partial \mathsf{C}_{avd,i}$ is the edge of $\mathsf{C}_{avd,i}$ and $t'_i$ is the intersection of the extension of the vector $v_i$ (i.e., focal agent $i$) with the repulsion area $\mathsf{SP}_{avd}$. The agent direction targets a point on the edge of this sphere cap (Fig 1D: lower).

## Boundary condition

We set no boundary condition, wall, or periodic condition since the periodic (or wall) boundaries do not accurately capture the interaction factors inside the flock due to periodic effects. The agents can move without constraint.

## Summary of algorithms

Fig 2 summarizes our algorithm. First, each agent checks if it has neighbors that must be repulsed. If so, an avoidance algorithm is applied. Following updating of all relevant agent positions according to the applied avoidance algorithm, a Delaunay triangle is constructed based on the current agent positions. Each agent computes the appropriate quasi-attraction and quasi-alignment. These two interactions determine the next direction of each agent. After all agent directions and positions are updated, one iteration is complete.

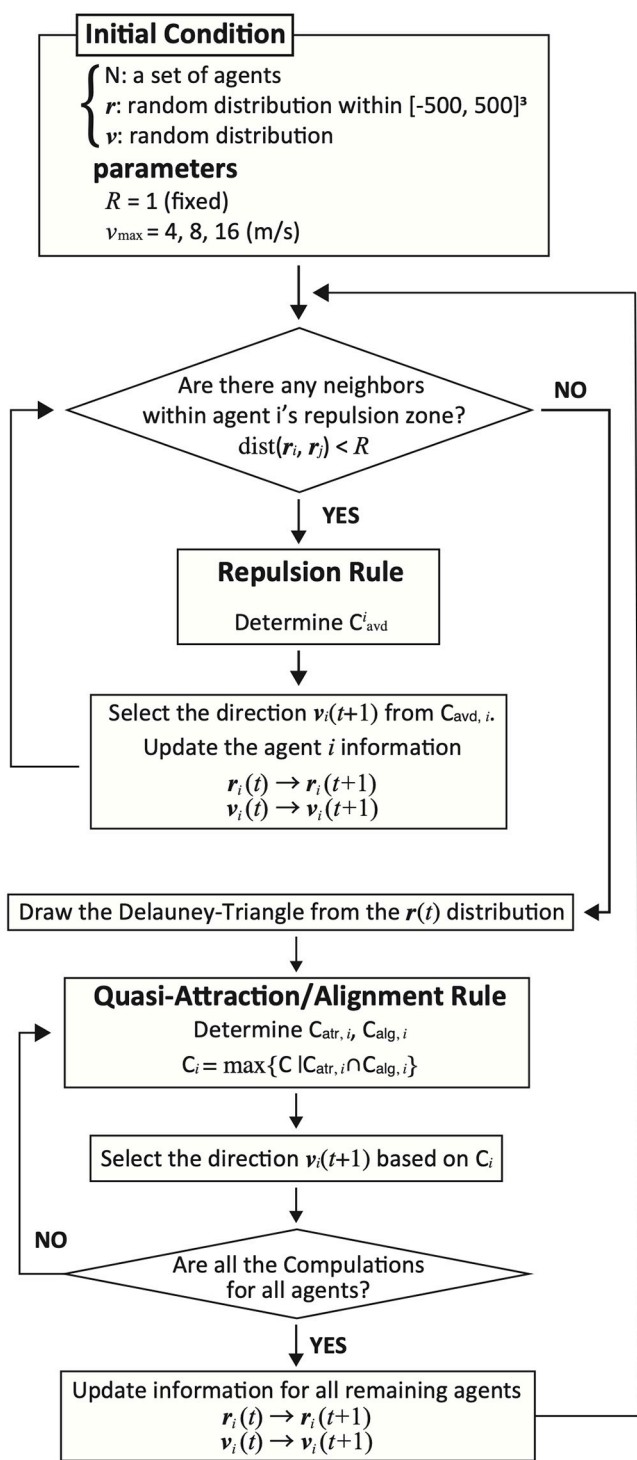

**Fig 2. Overview of algorithm.** One parameter (the maximum velocity $v_{max}$) exists when the repulsion radius $R$ is fixed. Each agent checks if there are any other individuals in the repulsion zone, and if not, it executes the quasi-attraction/alignment rule. The update timings are synchronous. The algorithm framework is of the same type as those of previous Boids models and other methods.

## Results

### Group formations

Our model uses only three parameters: number of agents, maximum speed, and repulsion size. We reduced the number of parameters to two as the degrees of attraction and alignment were self-tuned. Because only Delaunay triangles determined the agent positional relations, the topological relation was more important than the metric relation. Therefore, adjustment of the velocity–repulsion ratio $v_{\max}/R$ was sufficient to alter the group behavior.

In the simulations reported in this section, $v_{\max}/R = 8$ (this setting was based on the fish speed [24]) and the number of individuals was set to $N = 1000$. As already mentioned, the behavior of each agent varied dynamically according to the behaviors of its neighbors. This tendency indicates that the intrinsic noise of each agent could prevent perfect alignments and swarm-like noisy behavior.

Fig 3A confirms this characteristic. The group polarity $P$ (blue) was unstable for a given condition; in other words, it increased to 0.9 and then decreased to 0.5. The group considered when using our model dynamically changed formation through ambiguous interactions. In addition to this dynamic group formation (red), the group size $V_\alpha^{1/3}$ also varied ($V_\alpha$ is the volume of the group $\alpha$-shape [60], the definition of which is given in Section 1 in S1 Appendix). The volume fluctuation did not yield a uniform expansion, but rather, a heterogeneous one. The flock distorted in various ways (e.g., twists denoted in Fig 3C). Fig 3B shows the negative correlation between group skewness and $V_\alpha^{1/3}$ (correlation coefficient: −0.36). Group skewness

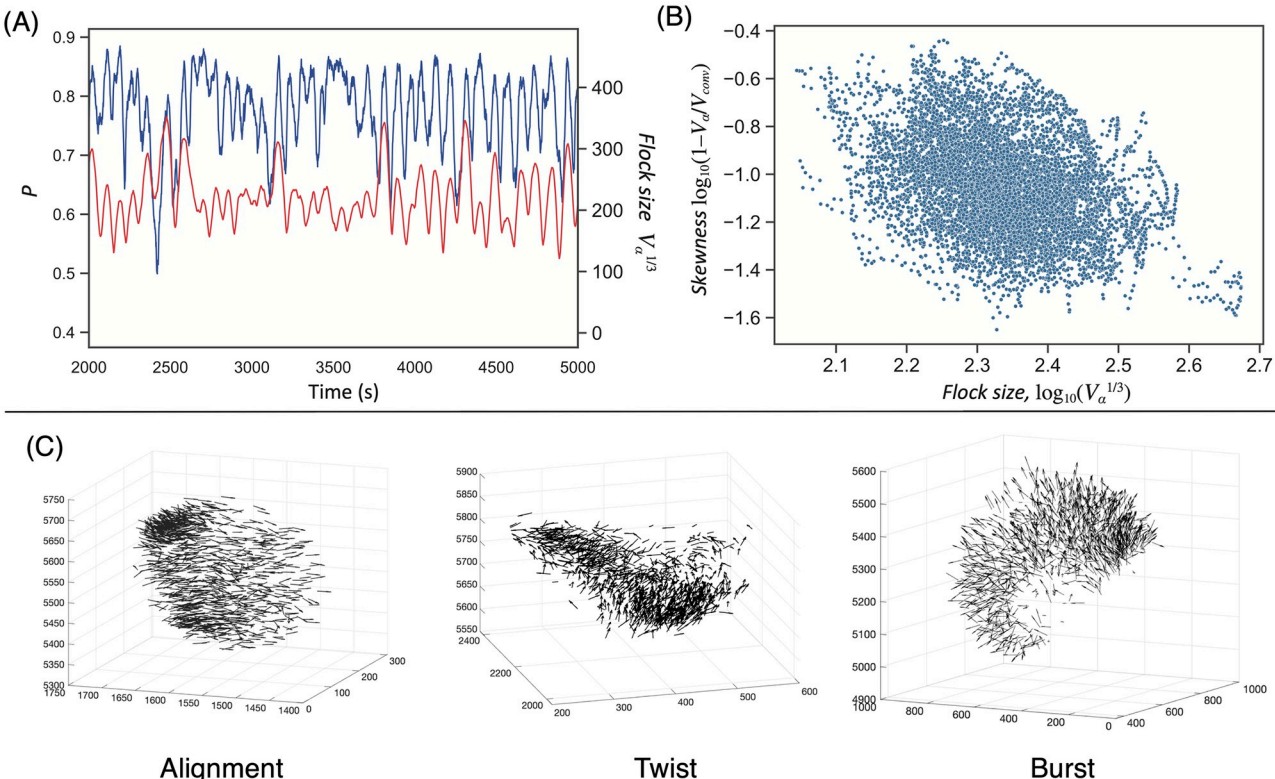

**Fig 3. Group formation.** (A) Sample time series for group polarity $P = |\sum_{i \in \mathbb{N}} \hat{v}_i|/N$ and flock size $V_\alpha^{\frac{1}{3}}$. (B) The negative correlation between $V_\alpha^{\frac{1}{3}}$ and its skewness $1 - V_\alpha/V_{conv}$, where $V_{conv}$ is the full convex volume (Pearson's correlation test: $n = 8000$, $r = -0.36$, $p < 10^{-30}$). Note that the inequality relation $V_\alpha < V_{conv}$ holds. (C) Examples of three types of group formation: alignment, twist, and burst.

can be defined as the ratio between the $\alpha$-shape volume and its full convex volume. If the $\alpha$-shape matches a full-convex formation, the skewness approaches zero. From the graph, the flock considered using our model tended to exhibit distorted shapes (such as twists or bursts, as shown in Fig 3C) when the value of $V_\alpha^{1/3}$ increased.

We identified three group formations often observed when using our model: (1) alignment: this formation is well-observed in all flocking models [37, 38]; (2) burst: this formation can be characterized as the one that occurs for low $P$ but large $V_\alpha^{1/3}$ and is often reported for fish schools [33, 61]; and (3) twist: this formation can be characterized as occurring for low $P$ but high local $P$ for bird flocks. From the outside, two groups in one collective interacted with each other. Sometimes, their directions were skewed; whereas the other times, they were directed opposite to one another, as if merging. We did not analytically examine these phenomena as it was sufficient to confirm that our model could successfully output various group formations from the given parameters.

## Lévy-walk and 1/$f$ fluctuation

First, we investigated the micro-level criticality, which does not pertain to the group, but rather, to the behavior of each individual.

We examined the internal trajectory of each agent for the center-of-mass reference frame. In our previous studies [24], we found that this type of a trajectory in a fish school indicates Lévy walks. We successfully replicated this result in the two-dimensional model [33]; however, it was unclear whether this finding extended to a higher dimension. Notably, a three-dimensional system has an additional degree of freedom than the two-dimensional case.

First, we briefly discuss Lévy walks observed in animal behavior. A Lévy walk is an optimal search strategy that balances exploration and exploitation [21–23]. It is a special type of a random walk and is in contrast to a Brownian walk, which exploits the surroundings of a specific location, as well as to a ballistic movement, which involves no exploitation and only exploration. The properties of the randomness of a Lévy walk lie between these two extremes. A Lévy walk is a characteristic property of animal behavior widely observed in nature [16–22]. Our previous study highlighted the possibility that a Lévy walk in a fish school facilitates efficient communication via super diffusion in the group [24].

Fig 4A shows an example of an agent trajectory for the center-of-mass reference frame. In this simulation, the trajectory spanned the entire group. In other words, each agent could have contact with neighbors in the flock. Fig 4B is an enlarged version of Fig 4A, where each dot on the line indicates the travel distance for one step. Here, the dot intervals were significant for the straight lines compared with the curved lines. This simple observation suggests that a Lévy-walk-like stop-and-go strategy may have been adopted.

To analyze this in greater detail, we applied the following method. We defined a "pause" (i.e., the zero-movement state) following the study conducted by Murakami and other researchers [24]. Thus, when $dr > ||\mathbf{r}_i(t) - \mathbf{r}_i(t-1)||$, or when the travel distance in the center-of-mass reference frame for one step was smaller than a given threshold, $dr$, agent $i$ was regarded to be pausing.

The rank distribution of the step length $l$ was found to obey a power-law distribution. More precisely, because the group size confined the individual agent trajectories, the graph was expected to follow a truncated power law, satisfying the following relation:

$$P(l) \sim \frac{1}{l_{min}^{1-\mu} - l_{max}^{1-\mu}} l^{-\mu} \tag{4}$$

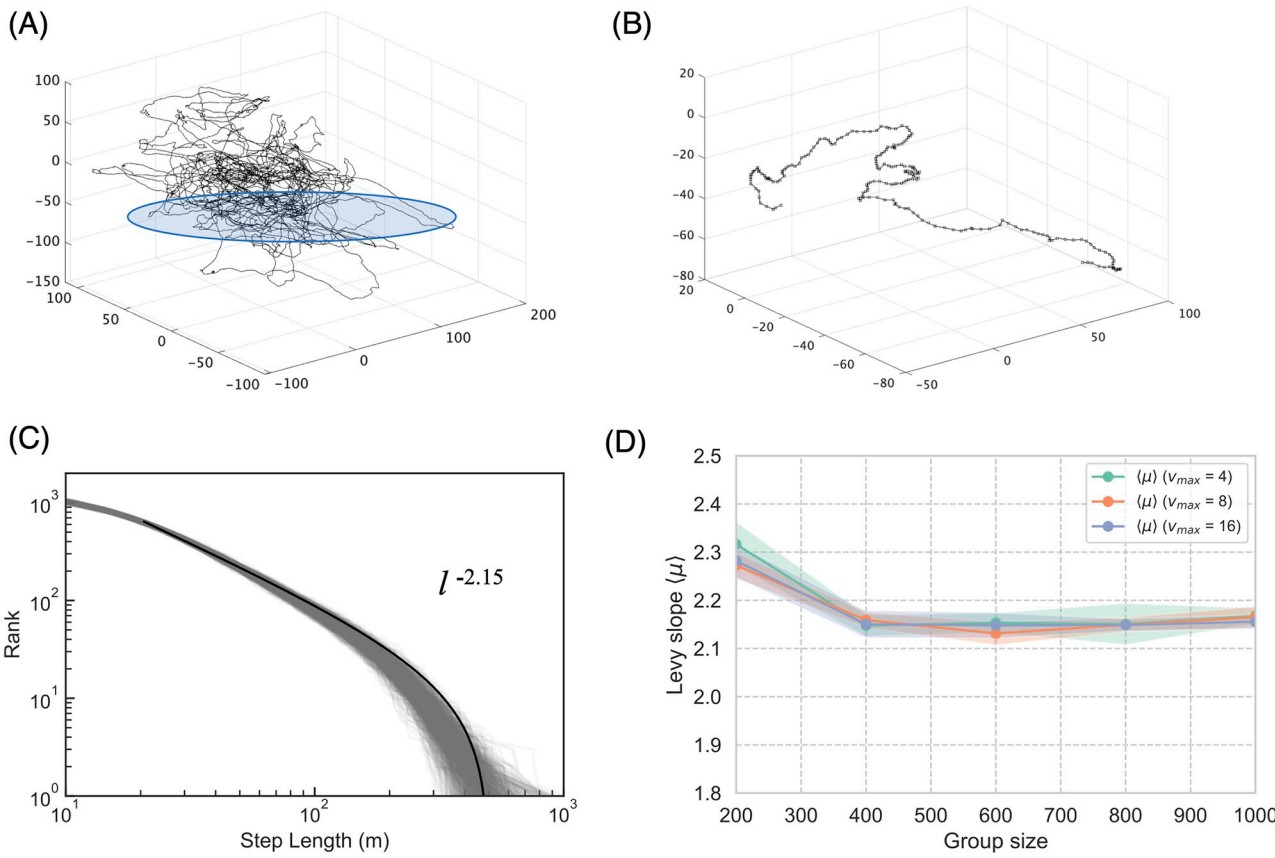

**Fig 4. Lévy walk on the center-of-mass reference frame.** (A) Sample center-of-mass trajectory for 10000 steps. (B) Enlarged version of the blue-circle area (200 steps). The dots indicate individual steps. (C) Sample step-length $l$ distribution for the mass-centered trajectories of 1000 agents. All $l$ distributions obey the truncated power law distribution $P(l) \sim l^{-\mu}$. The average Lévy slope $\mu$ is 2.15. (D) The average $\mu$ with respect to group size for various velocity $v_{\max}$ parameters.

where $l$ is a step length, where $\mu$ is the power law exponent, $l_{\min}$ is the minimum step length of a series of the trial, and $l_{\mathrm{mix}}$ is the maximum step length of a series of the trial.

If the $l$ distribution conformed to a Lévy walk, the exponential $\mu$ was approximately 2. Fig 4C shows a sample $l$ distribution for the mass-centered trajectories of 1000 agents. We observe that all the 1000 distributions were well-fitted with the truncated power law ($p > 0.10$ for the Kolmogorov–Smirnov test and the Akaike information criterion ($p$) = 1 compared with the fitted exponential [62]). Interestingly, the average slope $\mu$ was 2.15 (see Fig 4D). Our result suggests that all the agents in the same group performed Lévy walks, even in the three-dimensional space.

Additional results support the aforementioned finding. The jittering behavior for the center-of-mass reference frame also has critical statistical properties. Fig 5A shows the time series of velocity variation, $\|\boldsymbol{x}_i(t) - \boldsymbol{x}_i(t-1)\|$, of agent $i$ among 1000 agents, where $\boldsymbol{x}_i(t) = \boldsymbol{r}_i(t) - \boldsymbol{r}_{CM}(t)$ is the position of agent $i$ in the center of the reference frame, and $\boldsymbol{r}_i(t)$ is the position of agent $i$ at time $t$, $\boldsymbol{r}_{CM}(t)$ is the position of the flock center of mass at time $t$. Fig 5B shows the power spectrum of Fig 5A. The slope of this graph is approximately 1 for all the agents, which indicates $1/f$ fluctuation; this is also known as pink noise. When the slope is approximately 1, the time series fluctuation is said to be scale-free. In other words, a time series with strong self-similarity was obtained.

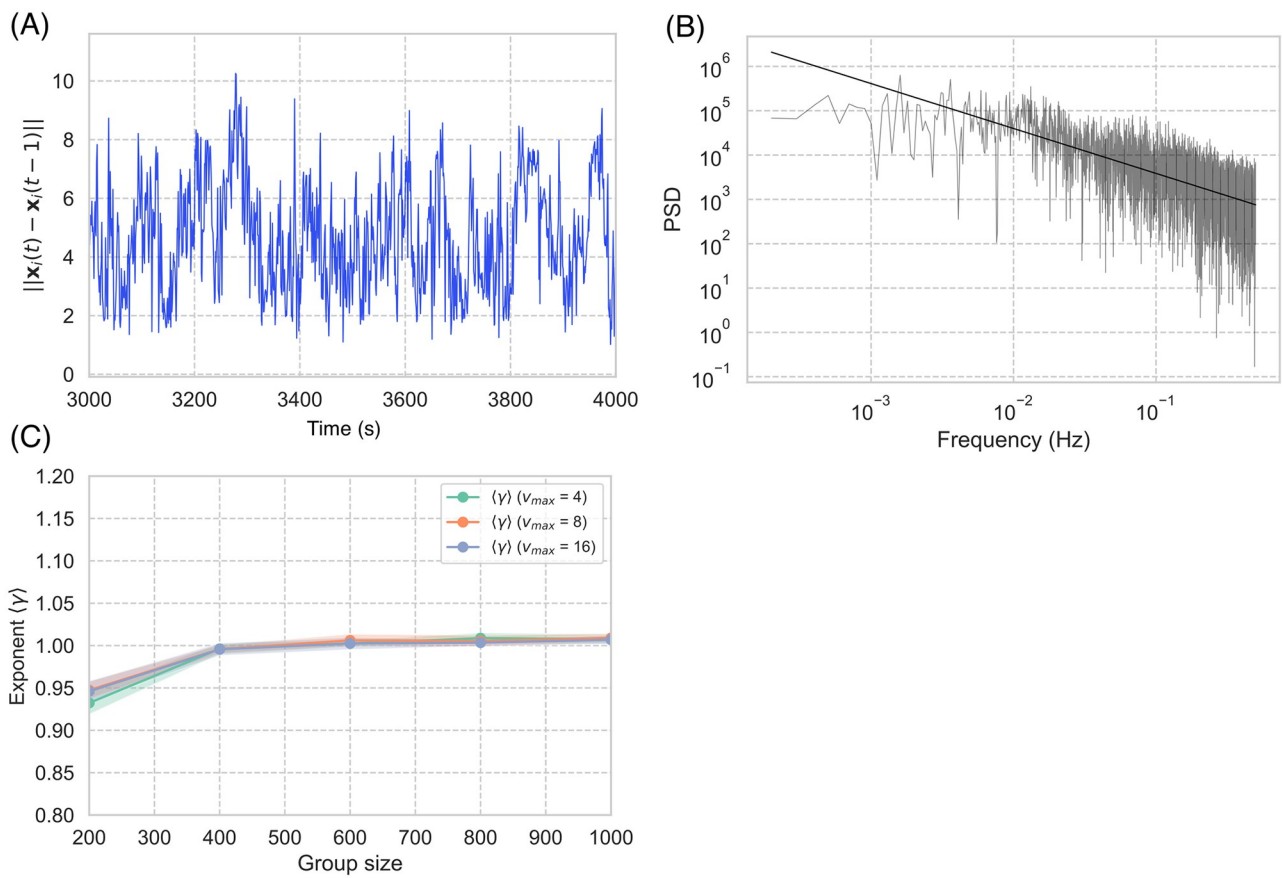

**Fig 5. Jittering behavior on center-of-mass reference frame.** (A) Sample velocity variation time series (i.e., $\|\boldsymbol{x}_i(t) - \boldsymbol{x}_i(t-1)\|$). (B) Power spectrum of (A): the slope of $f^{-\gamma}$ is 0.98 (solid line). (C) Average slope $\langle\gamma\rangle$ according to group size for different velocity $V$ parameters, with $\langle\gamma\rangle$ converging to 1.0. The jittering behavior of each agent is highly self-organized in time.

## Supper diffusion and scale free correlation

This subsection reports our examination of the macro-level criticality (i.e., the scale-free correlation). First, we examined the diffusive behavior of the flock. As some researchers have indicated, animal groups in nature exhibit high fluctuations (e.g., bird flocks [63], fish schools [12], and human crowds [25]). This instability in the intrinsic group behavior allows flocks to flexibly and dynamically respond to their dynamic environments [26, 27].

Super diffusion is an anomalous diffusion, which is high-speed compared with Brownian diffusion. To define this diffusion, the mean squared displacement must first be defined. Here,

$$\delta r^2(t) = \frac{1}{T-t}\frac{1}{N}\sum_{t_0=0}^{T-t-1}\sum_{i=1}^{N}[\boldsymbol{x}_i(t+t_0) - \boldsymbol{x}_i(t)]^2 \tag{4}$$

where $\boldsymbol{x}_i(t) = \boldsymbol{r}_i(t) - \boldsymbol{r}_{CM}(t)$ is the position of agent $i$ in the center of the reference frame, and $\boldsymbol{r}_i(t)$ is the position of agent $i$ at time $t$, $\boldsymbol{r}_{CM}(t)$ is the position of the flock center of mass at time $t$,. We averaged over all $N$ and overall time lags of duration $t \in [0, T]$. Here,

$$\delta r^2(t) = Dt^{\alpha} \tag{5}$$

and the behavior of the mean square displacement $\delta r^2(t)$ fit the power law, where $\alpha$ is the

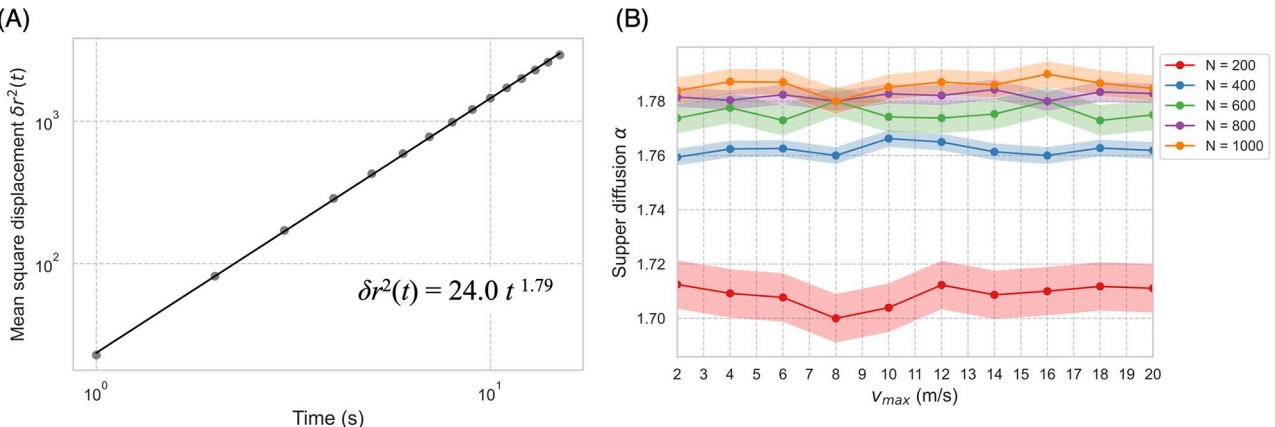

**Fig 6. Super diffusion.** (A) Mean square displacement in the center-of-mass reference frame. The diffusion coefficient $D = 24.0$, the diffusion exponent $\alpha = 1.79$, the group size $N = 1000$, and the velocity parameter $v_{max} = 8$ (m/s). (B) The $\alpha$ results obtained for different $v_{max}$ and $N$; hence, $\alpha$ depends on $N$ but not on $v_{max}$.

diffusion exponent from 0 to 2 and $D$ is the diffusion coefficient. If $\alpha$ was between 1 and 2, the system exhibited super diffusion [63].

Fig 6A shows that our model replicated super diffusion inside the flock (for mutual diffusion, see S1 Fig). In contrast with the empirical model, $\alpha$ was relatively high value around 1.78 for all parameter settings (Fig 6B). This high inner diffusion arose because no physical constraint was set on the model; that is, the condition for the *xyz* axis was homogeneous. However, actual flocks of starlings, for instance, have a high constraint on the *z*-axis because of gravity. In other words, movement in real flocks occurs in the medium region somewhere between two and three dimensions. The turning rate may have also impacted our result. In our model, the agents could change direction by up to $\pi/2$ radians (in the supplied Python code, this maximum turning rate can be tuned as a parameter S2 Appendix). Numerous flocking models adopt a maximal turning rate to replicate more realistic group behaviors. However, the aim of this study was not to mimic actual behaviors, but rather, to identify the origin of the criticality observed in nature while using minimal parameter settings. (Our model also behaved similarly to real flocks by exhibiting behaviors such as neighbor shuffling and boundary diffusion; see Section 5, 6, Fig Ca and Fig Cb in S1 Appendix).)

Next, we examined the scale-free correlation of the flock, which means that, even if the flock appears neatly aligned from the outside, the internal fluctuations (both orientation and speed) are not random but have a specific structure. Highly correlated fluctuation domains exist inside flocks and, in 2010, Cavagna et al. discovered that their domain sizes are scale-free [9]. Since Cavagna et al.'s discovery, numerous researchers have attempted to replicate their scale-free correlation results [14, 15, 40, 42]. Our interest is in confirming whether the high-fluctuation groups mentioned previously retain their scale-free properties.

To estimate the correlated domains inside a flock, we computed the correlation length $\xi$ at which the correlation function of the fluctuation became zero, i.e., $C(r = \xi) = 0$, using the relation

$$C(r) = \frac{1}{c_0} \frac{\sum_{i,j \in N} \boldsymbol{u}_i \cdot \boldsymbol{u}_j \delta(r - r_{ij})}{\sum_{i,j \in N} \delta(r - r_{ij})} \qquad (6)$$

Here, $\boldsymbol{u}_i$ is the fluctuation vector defined as $\boldsymbol{u}_i = \boldsymbol{v}_i - (\sum \boldsymbol{v}_i)/N$, $\delta(r - r_{ij})$ is the delta function, and $r_{ij} = |\boldsymbol{r}_j - \boldsymbol{r}_i|$ and $c_0$ are normalization parameters such that $C(r = 0) = 1$ holds. Further, $C(r = 0)$

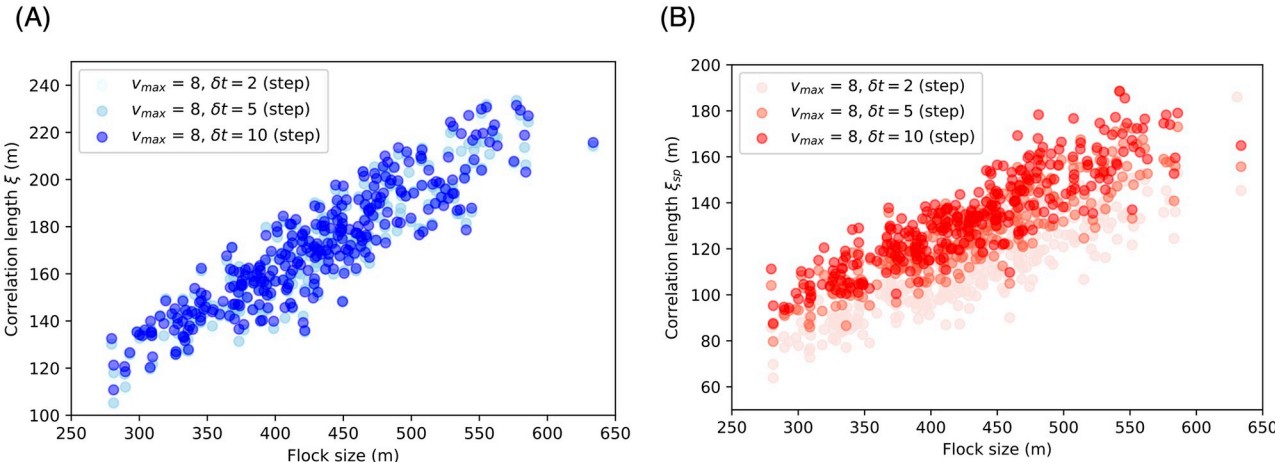

**Fig 7. Relation between correlation length and flock size.** The flock size is defined as the maximum distance between two agents. Instead of trajectory smoothing to cancel the agent noise, we examined the velocity vector $v = x^{t+\delta t} - x^t$ when $\delta t$ was 2, 5, and 10 steps. (A) The correlation between the correlation length $\xi$ and flock size $L$, where $\xi$ is proportional to $L$: $\xi = aL$ with $a = 0.40$ and $\delta t = 2$ (Pearson's correlation test: $n = 100$, $r = 0.93$, $p < 10^{-31}$), $a = 0.40$ and $\delta t = 5$ (Pearson's correlation test: $n = 100$, $r = 0.93$, $p < 10^{-31}$), and $a = 0.40$ and $\delta t = 10$ (Pearson's correlation test: $n = 100$, $r = 0.93$, $p < 10^{-31}$). (B) The correlation between the correlation length $\xi_{sp}$ and $L$, where $\xi_{sp}$ is proportional to $L$: $\xi_{sp} = aL$ with $a = 0.26$ and $\delta t = 2$ (Pearson's correlation test: $n = 100$, $r = 0.92$, $p < 10^{-31}$), $a = 0.30$ and $\delta t = 5$ (Pearson's correlation test: $n = 100$, $r = 0.92$, $p < 10^{-31}$), and $a = 0.31$ and $\delta t = 10$ (Pearson's correlation test: $n = 100$, $r = 0.92$, $p < 10^{-31}$). For the other parameter settings, see S1 Table.

is 1 and gradually decays to negative values along with $r$. We computed the correlation function for the speed fluctuation in a similar manner (see Section 7 in S1 Appendix).

Fig 7A and 7B show that high scale-free properties were also observed for our highly fluctuated model, especially as regards their orientation. The orientation slope exceeded 0.40. In contrast, the speed slope was low (about 0.31) compared with available data for starlings [9]. This result may have been caused by our velocity algorithm. (A recent study has suggested another approach to coupling the velocity function with the restoring forces [64].) Nevertheless, our result indicates that both cases exhibited scale-free correlation.

The correlated domain in the group could be observed graphically. Fluctuations within the group could be divided into two groups (red and blue in S4 Fig). The next subsection reports our use of this property for analyzing information transfer between the two regions.

### Information transfer between correlated subgroups

**Constructing scale-free induced subgroups.** Thus far, we have investigated nested criticality in a group based on ambiguous interactions. This subsection reports our investigation of information flow in terms of critical properties.

To obtain a clearer information flow inside the flock, we divided the group into two subgroups using the scale-free correlation. Although we confirmed the existence of two highly correlated groups by their criticality, construction of a clear-cut criticality for the group using fluctuations alone was challenging. If scale-free correlations were used alone, some agents may have been mixed with opposite fluctuation groups. Without clear group divisions, we could not define the information flow among the subgroups. Therefore, additional information was required to construct distinct subgroups.

We applied the $k$-means method, which is a clustering method for given vector information, to construct two subgroups. We used the $k$-means function in Matlab (MathWorks Inc., Natick, USA), taking the position and unit fluctuation vectors as inputs and setting the division number (parameter $k$) to 2. As $k$-means clustering does not provide a unique division, we

---

**Algorithm** : Scale-free induced subgroup

**Data:** $\mathbf{r}, \hat{\mathbf{u}}$

; /* r:position vectors, û:unit fluctuation vectors */

**Result:** Index

**for** $t \leftarrow 1$ **to** 20 **do**

 J = kmeans($[\mathbf{r}, \hat{\mathbf{u}}], k = 2$);

 **for** $i \leftarrow 1$ **to** $N$ **do**

 **if** J($i$) $== 1$ **then**

 count($i$)$+ = 1$;

 **end**

 **end**

**end**

; /* Determine each index: group $i = 1$ or $i = 2$ */

**for** $i \leftarrow 1$ **to** $N$ **do**

 **if** count($i$) $\geq 10$ **then**

 Index($i$) $= 1$;

 **else**

 Index($i$) $= 2$;

 **end**

**end**

---

**Fig 8. Algorithm of the scale-free induced subgroup.** The input information is $\{\mathbf{r}_i\}_{i \in \mathsf{N}}$ and $\{\hat{\boldsymbol{u}}_i\}_{i \in \mathsf{N}}$ (i.e., the fluctuation positions and directional information). The output index is the assignment group number.

performed the same computation 20 times and selected the largest belonging member number in each case (see count($i$) in Fig 8). Finally, we reindexed each group according to the alignment order in the mean direction: the top group was indexed as the "leader" and those behind were indexed as "followers." S4 Fig shows that grouping based on the scale-free correlation yielded distinct divisions. We note that the number of subgroups was stable (the mean subgroup size was approximately 500 and the survival probability of the same subgroup decayed at a considerably slower degree than an exponential decay). More than 70% of the subgroup remained unchanged after 50 steps (S5 Fig). This stability confirms that our method successfully reflected the fluctuation vector property.

**Mutual information for two input types.** We measured the information flow between two subgroups induced by scale-free correlation, considering the two types of average vectors. In other words, we prepared two patterns of interaction between a pair of average vectors for each reindexed group (i.e., two average velocity vectors $\{\langle V \rangle_{\text{leader}}, \langle V \rangle_{\text{follower}}\}$ and two average fluctuation vectors $\{\langle \boldsymbol{F} \rangle_{\text{leader}}, \langle \boldsymbol{F} \rangle_{\text{follower}}\}$). This coarse-graining view elucidated the interaction between the two groups as an interaction between two virtual representative agents. The effect of these vectors could be evaluated as the future behavior of the group. We measured the behavior of this group as the degree of curvature $K$ of the entire trajectory. We could compute the curvature vectors $\boldsymbol{k}$ at each point from the three position vectors of the center-of-mass trajectory of the flock (Fig 9A). We obtained a certain interval for the average to cancel its local zig-zag trajectories; in other words, $K(t) = |\sum_t^{t+\delta t} \boldsymbol{k}(t)|/\delta t$. Therefore, the effects of the two

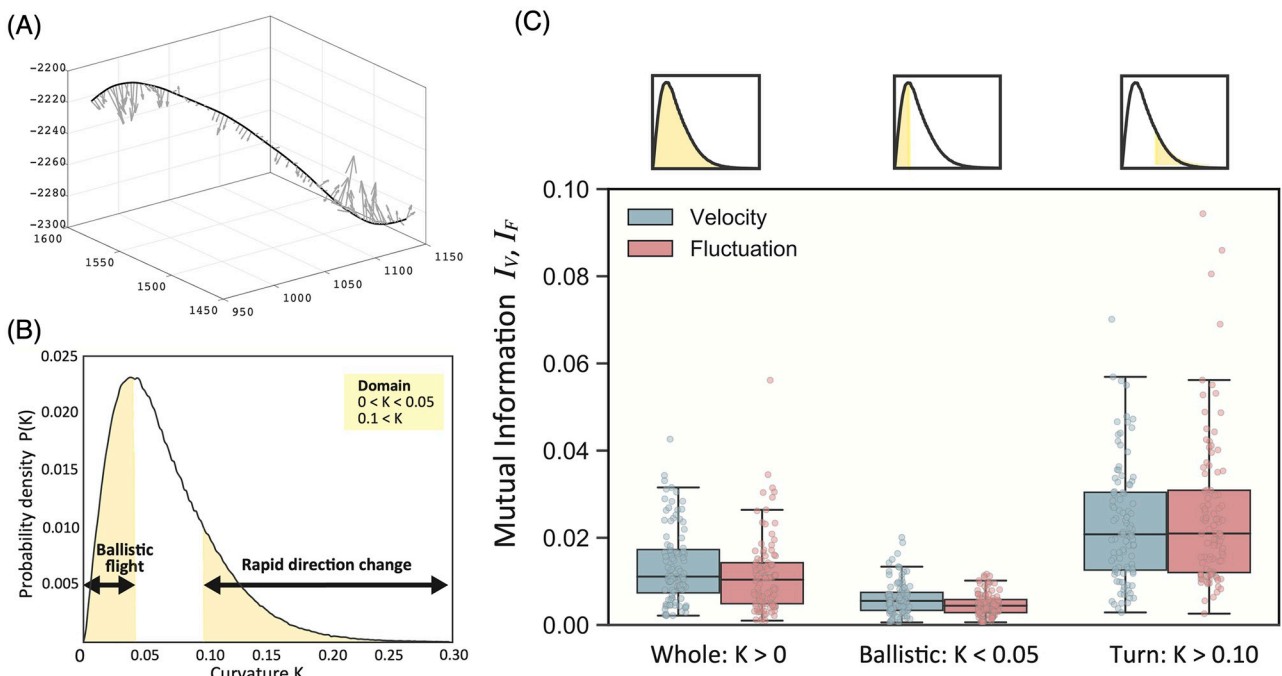

**Fig 9. Curvature vectors and mutual information obtained for different inputs.** (A) The flock center-of-mass trajectory and its curvature vectors at each point. (B) The probability density $P(K)$ for all series of data where $\delta t = 5$ steps. We defined the region with curvature $K < 0.05$ as the ballistic region, and the region for which $K > 0.10$ as the group turning region. (C) Mutual information $I_V \equiv I(\{V_{\text{leader}}(t), V_{\text{follower}}(t)\}; K(t))$ and $I_F \equiv I(\{F_{\text{leader}}(t), F_{\text{follower}}(t)\}; K(t))$ for each region. The mutual information (i.e., $I_V$, $I_F$) in the group turning region is significantly larger than in the ballistic region. The detailed statistical test results are listed in S1 Table.

vectors could be computed as mutual information: $I(\{V_{\text{leader}}(t), V_{\text{follower}}(t)\}; K(t))$ or $I(\{F_{\text{leader}}(t), F_{\text{follower}}(t)\}; K(t))$.

Fig 9B shows the frequency distribution of $K(t)$ extracted from a series of flocks (1000 agents). In the low-$K(t)$ region, the flock exhibited ballistic movement. In contrast, in the high-$K(t)$ region of Fig 9A, significant directional changes occurred. To understand the effect of information flow on flock turning, we identified two regions for the analysis: ballistic regions and turning regions, i.e., those with $K(t) < 0.05$ and $K(t) > 0.10$, respectively.

Fig 9C shows the mutual information from 100 trials of the velocity (blue) and fluctuation (red) vectors. Three patterns were identified for the analysis: whole, ballistic, and turning. The turning condition featured more mutual information than the other two conditions. In particular, relatively little mutual information was required for ballistic movement. This observation indicates that the velocity and fluctuation information transfer between the subgroups is significantly related to changes in the whole-flock turning behavior. Next, we focused on high-$K(t)$ ($> 0.1$) regions only, as their mutual information was sufficiently large for the application of PID.

**Application of PID to rapid group turning.** To understand the turning condition in more detail, we applied PID to the given mutual information. PID ensures that any mutual information can be decomposed into four elements: redundancy, two unidirectional information flows, and synergy [34, 35, 65, 66]. The use of transfer entropy to measure the information flow between two systems has been criticized [67–69] as over- or under-estimation may occur because the redundancy and synergy are omitted [67, 70]. To elucidate the information flow between two groups, the transfer entropy may be insufficient as the pure information transfer from one side to another must be estimated. Thus, the redundancy and synergy effects in the

system must be considered. In our analysis, we used the PID Matlab code given in [66] (https://github.com/robince/partial-info-decomp).

The PID equation for the two inputs is given as follows:

$$I(\{\boldsymbol{X}_1(t), \boldsymbol{X}_2(t)\}; K(t)) = R(t) + U_1(t) + U_2(t) + S(t) \tag{7}$$

where $R(t)$ is the redundancy, $U_1(t)$ is the unique information flow from $\boldsymbol{X}_1(t)$ to $K(t)$, $U_2(t)$ is the unique information flow from $\boldsymbol{X}_2(t)$ to $K(t)$, and $S(t)$ is the synergy.

For an easy understanding of the PID, a simple binary logic circuit is suitable (we present a figurative interpretation in S6 Fig; alternatively, see [69]). However, this interpretation is valid for discrete inputs only. Thus, we briefly describe PID here for readers unfamiliar with this approach.

Note that redundancy increases when the two given inputs (i.e., $X$, $Y$) are correlated. This explains the use of the term "redundancy" as when two inputs are highly correlated, the output (i.e., $Z$) contains unnecessary information and the system is highly redundant. The lack of one input does not affect the system output in terms of redundancy. If we subtract the redundancy $R$ from $I(X; Z)$, we obtain a unique information flow $U_1$. Compared to transfer entropy, this quantity can more accurately measure the flow of information. The synergy is defined as the remainder obtained when subtracting $R$, $U_1$, and $U_2$ from $I(X, Y; Z)$. In contrast with redundancy, this quantity measures the non-linear effect generated by the combination of two inputs. One input cannot generate the same output as the other; i.e., redundancy and synergy play different input roles. The input relation of redundancy is symmetric as each input has the same function as regards the output, whereas the input relation of the synergy is asymmetric as two inputs act as a set for the output.

We applied the PID to the rapid group turning region (i.e., $K(t) > 0.10$). According to our settings, if the information flow was unidirectional from the leader to the follower, $U_1(\{\boldsymbol{V}_{\mathrm{lead}}(t)\}; K(t))$ was the largest and the other three were smaller; however, Fig 10A shows a different result. For the velocity inputs, the largest was the synergy information $S(\{\boldsymbol{V}_{\mathrm{lead}}(t), \boldsymbol{V}_{\mathrm{follow}}(t)\}; K(t))$ and the second-largest was the redundancy $R(\{\boldsymbol{V}_{\mathrm{lead}}(t), \boldsymbol{V}_{\mathrm{follow}}(t)\}; K(t))$ (the same

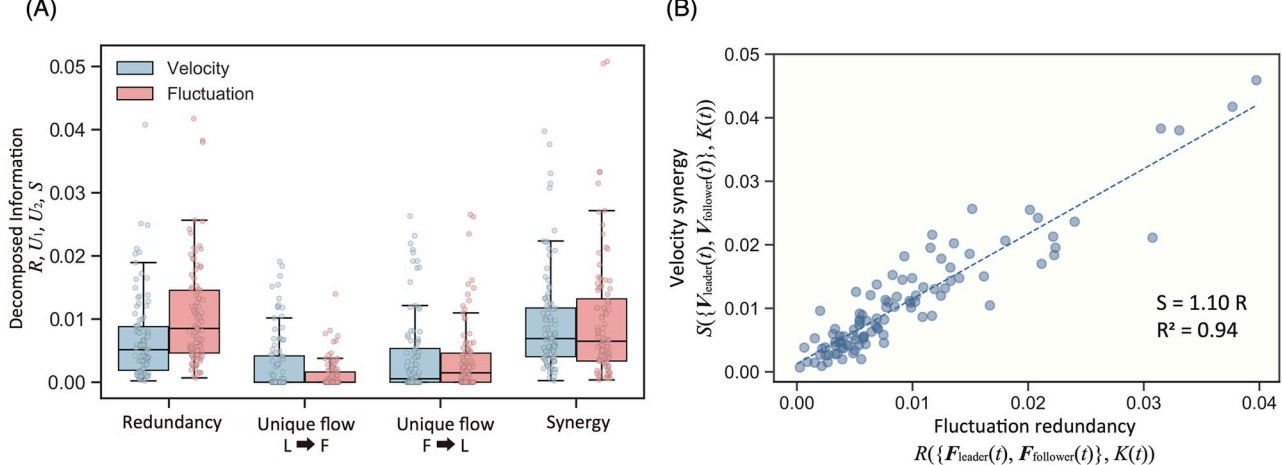

**Fig 10. PID for group turning phase.** (A) The PID decomposition for $I_V$ (blue) and $I_F$ (red) of the turning result in Fig 9C. The velocity synergy is higher than that of the redundancy (paired $t$-test: $t(99) = 2.75$, $p < 0.01$); however, the fluctuation synergy does not exceed the fluctuation redundancy (paired $t$-test: $t(99) = -0.934$, $p > 0.1$). The other statistical test results are listed in S1 Table. (B) The relation between the velocity synergy $S(\{\boldsymbol{V}_{lead}(t), \boldsymbol{V}_{follow}(t)\}; K(t))$ and fluctuation redundancy $R(\{\boldsymbol{F}_{lead}(t), \boldsymbol{F}_{follow}(t)\}; K(t))$. The correlation efficient is 0.93 (Pearson's correlation test: $n = 100$, $p < 10^{-31}$). The other parameter settings are reported in S1 Table.

degree for the fluctuation inputs). The two unidirectional flows for both cases were relatively low. For the velocity inputs, $U_1(\{V_{\text{lead}}(t)\}; K(t))$ was larger than $U_2(\{V_{\text{follow}}(t)\}; K(t))$; however, both the values were small. Our result suggests that the group information transfer did not reflect the leader–follower relation. In other words, it was confirmed that positional information is not a crucial factor in group turning.

By now, it is clear how the velocity and fluctuation inputs affected group turning, and we also examined the relationship between the two inputs. Fig 10B indicates a high correlation between the synergy of the velocity input $S(\{V_{\text{lead}}(t), V_{\text{follow}}(t)\}; K(t))$ and the redundancy of the fluctuation input $R(\{F_{\text{lead}}(t), F_{\text{follow}}(t)\}; K(t))$. We discuss the significance of this relationship in more detail in the Discussion section. Here, however, we note only the correlation. That is, the group turning couples the asymmetry of the velocity inputs to the symmetry of the fluctuation inputs. This dual aspect effectively reflects the critical nature of the system, which we identified throughout this study. The system maintains its high susceptibility to external stimulus from the fluctuation perspective; from the velocity perspective, our findings suggest that the system responds to some non-linear effect. Interestingly, the inverse relationship does not hold (S1 Table). The different roles of the velocity and fluctuation inputs may be an essential property of the system criticality.

## Discussion

In this study, we demonstrated the nested criticality that emerged from a single algorithm employed our proposed ambiguous interaction model, which is a natural extension of the representative flocking model. We re-interpreted the attraction and alignment as short- and long-term predictions, respectively, and as a key concept of our model, we considered these predictions as regions (i.e. $\mathbf{C}$) rather than points. This vagueness offered the agents with several options in specific contexts. As our model does not contradict the concepts of the Boids model or other models, appropriate statistical properties could be confirmed (e.g., scale-free correlation and super diffusion).

We also confirmed several micro-criticalities (i.e., Lévy walks and $1/f$ fluctuation) caused by self-tuning noise. In our model, the next direction of the focal agent is provided as a region $\mathbf{C}$; thus, the behavior of that agent strongly depends on the behavior of its neighbors. If $\mathbf{C}$ decreases or increases (e.g., with local high/low polarity), the next direction accordingly becomes deterministic or indeterministic. This continuous oscillation between deterministic and indeterministic behavior yields the criticality of each individual. Furthermore, the two observed criticalities are in stark contrast. While the Lévy walk relates to the criticality in space, the $1/f$ fluctuation is related to the criticality in time. Some researchers suggest that the self-similar structure of a time series for jittering behavior is related to the Lévy-walk result [71–73]. Although it is uncertain whether our result can be applied to actual data, we will confirm this relationship in future research.

Before discussing the group turning behavior, we consider the meaning of the micro-scale criticality, which pertains to system robustness. Generally, robustness is quite a different concept to stability [74]. System stability can be described as its "persistence" in the original state, whereas system robustness relates to the "interplay between organization and dynamics" [74]. There is no original state to which the flock can return. In this sense, the flock is not a stable system; however, it is robust, as its shape and behavior change dynamically depending on the situation. In fact, some studies have suggested that robust systems should include micro-level criticality. For example, Ros et al. showed that, even for simple models such as the self-propelled particle model, various collective behaviors can emerge through the application of the logistic map update rule at Feigenbaum critical points [75]. Another example is the inverse

Bayesian inference [76]. Unlike Bayesian inference, inverse Bayesian inference involves permanently changing a hypothesis, and Gunji et al. successfully reproduced Lévy walks and various collective phenomena by applying Bayesian and inverse Bayesian inference [76–78]. All these studies have shown that micro-level criticality constitutes the robustness of group behavior. The nested criticality confirmed in this study is also consistent with these findings.

Next, we considered the functional role of criticality. Although the scale-free correlation may relate to group turning behavior, the lack of experimental data has prevented further investigations. However, our scale-free induced coarse-gain group method elucidated the relationship between criticality and group turning behavior. Our method assumes a macro-criticality that is stable for a certain interval. Using this approach, we showed that the interactions between correlated domains contribute to rapid group turning involving both velocity and fluctuation distributions.

The PID analysis conducted in this study provided us with a more detailed description of this mechanism. The PID uniquely decomposed the mutual information $I(X, Y; Z)$ into four types of information: redundancy, two unique information flows, and synergy (where $X$ and $Y$ are inputs and $Z$ is the output). Recall that redundancy means that the system contains a compensatory input for an output. In other words, if one input (i.e., $X$) is missing, the same output can be expected from the other source as well (i.e., $Y$). In the case of continuous inputs, redundancy refers to correlational inputs. As inputs are symmetrical in such cases, one side of the information can be recovered from the other. The unique information flow is the remainder of the mutual information; that is, $I(X; Z)$ (or $I(Y; Z)$) minus the redundancy. This quantity resembles the transfer entropy, but we must not confuse the two as the transfer entropy sometimes over- or under-estimates the net information [67]. The unique information flow complements this disadvantage of transfer entropy. Finally, synergy refers to the remainder of the mutual information, i.e., $I(X, Y; Z)$ minus the two unique information flows and redundancy. In contrast to redundancy, the pair of synergy inputs contribute to the output. The effect of the inputs on the output is non-linear. In this case, the loss of one source of information cannot reproduce the same output.

The above-mentioned discussion explains why high redundancy emerges in the fluctuation inputs, $R(\{F_{\text{lead}}(t), F_{\text{follow}}(t)\}; K(t))$. The strong correlation between the two groups (i.e., $F_{\text{lead}}$ and $F_{\text{follow}}$) comes from the scale-free correlation. This input symmetry of $F_{\text{lead}}$ and $F_{\text{follow}}$ also resonates with the critical-system susceptibility. The flock certainly contributes to the group turning by increasing its susceptibility. In contrast, the synergy for the velocity inputs, $S(\{V_{\text{lead}}(t), V_{\text{follow}}(t)\}; K(t))$, is more substantial than its redundancy (paired $t$-test; $t(99) = -2.76$, $p < 0.001$). This tendency means the asymmetrical input relation of $V_{\text{lead}}$ and $V_{\text{follow}}$ is the key to the group turning behavior for the velocity distribution. Our analysis suggests that the velocity and fluctuation distributions contribute to the group turning according to their different roles.

The strong correlation between $S(\{V_{\text{lead}}(t), V_{\text{follow}}(t)\}; K(t))$ and $R(\{F_{\text{lead}}(t), F_{\text{follow}}(t)\}; K(t))$ provides a more detailed picture. No other combination showed such a strong relationship, which suggests a structural coupling. Although the influence of $R(\{F_{\text{lead}}(t), F_{\text{follow}}(t)\}; K(t))$ means that the correlated fluctuation inputs contribute to the group turning, the meaning of $S(\{V_{\text{lead}}(t), V_{\text{follow}}(t)\}; K(t))$ remains unclear. Intuitively, the two subgroups contribute independently (i.e. non-correlated) to the group turning. Considering this asymmetric input relation, the velocity vector distribution should be considered here. We call this type of velocity distribution "torsion," as the two velocity inputs exhibit high torque during the rapid group turning (see S1 Table; alternatively, the distorted formation during group turning is visible in S1 Video). Considering the torque as object rotation, the high torque velocity distribution can relate to the group turn.

Therefore, the correlation between $S(\{\boldsymbol{V}_{\text{lead}}(t), \boldsymbol{V}_{\text{follow}}(t)\}; K(t))$ and $R(\{\boldsymbol{F}_{\text{lead}}(t), \boldsymbol{F}_{\text{follow}}(t)\}; K(t))$ should be interpreted as follows: A high fluctuation correlation and an appropriate flock morphological structure (i.e., velocity torsion) are needed to generate a rapid group turn. Thus, a suitable morphology coupled with correlated fluctuations to transform the fluctuation power are required for group turning behavior.

This proposition also indicates that the inverse relation (i.e., between $S(\{\boldsymbol{F}_{\text{lead}}(t), \boldsymbol{F}_{\text{follow}}(t)\}; K(t))$ and $R(\{\boldsymbol{V}_{\text{lead}}(t), \boldsymbol{V}_{\text{follow}}(t)\}; K(t))$) does not hold. High $R(\{\boldsymbol{V}_{\text{lead}}(t), \boldsymbol{V}_{\text{follow}}(t)\}; K(t))$ yields a symmetric distribution of the velocity vectors only, whereas high $S(\{\boldsymbol{F}_{\text{lead}}(t), \boldsymbol{F}_{\text{follow}}(t)\}; K(t))$ indicates low $R(\{\boldsymbol{F}_{\text{lead}}(t), \boldsymbol{F}_{\text{follow}}(t)\}; K(t))$ (Pearson's correlation test: $n = 100$, $r = -0.92$, $p < 10^{-40}$). Therefore, the inverse relation states that the heterogeneous fluctuation distribution has no relation with the homogeneous velocity distribution in terms of generating a group turn. The coupling with the fluctuation power may be disconnected when the sub-groups work as correlational units.

In this study, we considered only one parameter, $v_{\text{max}}/R$. It remains to be seen whether the same results can be obtained by adding more realistic constraints (e.g., a maximum turning rate, visual field, and gravity effect) to our model. However, the behavior of our model is statistically close to those of real flocks. We also noted that the realistic conditions implemented using the Boids model cannot replicate a more accurate description of group criticality [42]. Furthermore, few models focus on criticality across several levels. The effect of further adding more realistic assumptions to our model remains to be investigated in a future study.

Our flock model suggests that the nested criticality within the flock has dual effects on the group turning behavior, via the fluctuation power in the form of symmetric fluctuation inputs and via the group morphology in the form of asymmetric velocity inputs. Both the aspects have functional roles when coupled. The flock morphology may help its internal fluctuation convert to dynamic behavior. However, several studies have paid a significant amount of attention to the former but not to the latter [9, 28–32, 40, 79, 80]. This study therefore suggested the importance of group morphology and that group criticality may support such couplings. The functional role of group criticality described in this study is a verifiable hypothesis if sufficient data become available.

## Supporting information

**S1 Fig. Mutual diffusion.**
(PDF)

**S2 Fig. Border diffusion for other parameter settings.**
(PDF)

**S3 Fig. Scale-free correlation for other parameter settings.**
(PDF)

**S4 Fig. Sample correlated domains inside the flock.** Each color indicates a highly correlated sub-flock inside the overall flock using the $k$-means method.
(PDF)

**S5 Fig. Sample survival probability within the same subgroup.**
(PDF)

**S6 Fig. Figurative PID image.**
(PDF)

**S1 Video. Sample of flocking behavior ($N = 600$ and $v_{max} = 8$).**
(MOV)

**S1 Appendix. Supporting information concerning the algorithm, its nomenclature, and other definitions.**
(PDF)

**S2 Appendix. Listed python codes of our algorithm with explanations (Python 3.9 recommended).**
(RAR)

**S1 Table. Statistical test results for other parameter settings.**
(XLSX)

## Author Contributions

**Conceptualization:** Takayuki Niizato, Hisashi Murakami.

**Data curation:** Takayuki Niizato.

**Formal analysis:** Takayuki Niizato.

**Funding acquisition:** Takayuki Niizato.

**Investigation:** Takayuki Niizato.

**Methodology:** Takayuki Niizato, Takuya Musha.

**Project administration:** Takayuki Niizato.

**Resources:** Takayuki Niizato.

**Software:** Takayuki Niizato.

**Supervision:** Takayuki Niizato.

**Validation:** Takayuki Niizato.

**Visualization:** Takayuki Niizato, Takuya Musha.

**Writing – original draft:** Takayuki Niizato.

**Writing – review & editing:** Takayuki Niizato, Hisashi Murakami.

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
