## [Decision Letter · Decision Letter 0]

14 Sep 2022

Dear Dr. Niizato,

Thank you very much for submitting your manuscript "Functional duality in group criticality via ambiguous interactions" for consideration at PLOS Computational Biology.

As with all papers reviewed by the journal, your manuscript was reviewed by members of the editorial board and by several independent reviewers. In light of the reviews (below this email), we would like to invite the resubmission of a significantly-revised version that takes into account the reviewers' comments.

In light of the three reviews, we would like to ask you to thoroughly address their comments and suggestions, especially R#1's. We look forward to receiving your revised manuscript.

We cannot make any decision about publication until we have seen the revised manuscript and your response to the reviewers' comments. Your revised manuscript is also likely to be sent to reviewers for further evaluation.

Sincerely,

Xingru Chen, Ph.D.

Guest Editor

PLOS Computational Biology

Natalia Komarova

Section Editor

PLOS Computational Biology

In light of the three reviews, we would like to ask you to thoroughly address their comments and suggestions, especially R#1's. We look forward to receiving your revised manuscript.

Reviewer's Responses to Questions

**Comments to the Authors:**

Reviewer #1: This paper presents a new model of collective animal behavior. The model is a boid type model which relies on spheres of influence. Instead of just having the normal terms of repulsion, alignment and attarction, the authors show how alignment and attarction are the same process but at different time scales. As such they can use intermediate versions in their model. The model introduces noise as an intrinsic parameter to the agents of the model, based on the sphres of influence form its neighbors, instead of an added tuned parameter like in many other models.

They then procede to show how this model produces critical behvaior in many different ways as observed in other studies. They do not attempt to re-create exact animal behaviors, but rather a simple model that can reproduce the observed phenomena.

I recomend to reject this article, as I find the writing hard to follow, and the detail of the methods applied lacking.

There is a general lack of articles (an, a the) throughout the paper. Spelling mistakes such as supper instead of super.

It is difficult to understand parts of the manuscript as lines 130-134, line 144, line 200 (the von distribution ??),

For the methods, some quantities, e.g. mu only appears on line 201, then not until the caption of Figure 7. Similarly for kappa, which only appears on line 201, without describing what it is.

Combinging this writing with the very compact mathematical notation for the model, such as equation 2 and 3 along with the long notation in the explanation of these equations makes the paper difficult to understand.

It also does not appear from the paper how a Partial Information decomposition is done.

In a similar vein it would be nice with just a brief mention of how the k-means partitioning of the flock into two subgroups was preformed, to make this a stand-alone piece of research.

There is new information presented in the discussion, such as line 551: ''The agents of our model have intact visual fields,''.

The conclusion is also extremely short, and does not provide much infomation on the exact value of this model over other models.

It would have been nice with a clearer statment on which aspects this model captures better, or if it fits more accurately to observational data (I know the authors explicitly says they do not compare with real data, but this would lift the impact of this manuscript.), of it this new model captures mode different critical behaviors than other models of flocking.

Reviewer #2: This manuscripts examines a clever extension of the theory of flock dynamics and finds that many collective behaviors can be deduced from the extension proposed. The manuscript presents, a somewhat complicated version, of the alignment and attraction model used. The main point of the manuscript can be found (at last!) in the following line that explains an interesting point: What is alignment? The authors present an interesting extension of this concept that has some overtones with the concept of Anticipation.

Or in line 132

“In these contexts, our claim is clear: both interactions stem from a common interaction. “

“We showed that he interaction between alignment and attraction, having been applied to many flocking models, can be regarded as a single interaction in terms of time-scale differences. The alignment indicates the infinity long predicted position of the neighbors, whereas attraction indicates the interaction with the current position of the neighbors. We showed that the medium region of this time scale could play a critical role in group criticality. However, our result is restricted to the two-dimensional condition.”

The conclution are interesting …and could be generalized to other situations different from flock formation.

The papers suffers from some (resolvable) problems in the exposition.

1) a very vert minor point …Dr. Niizato does not have an institutional email? (t_niizato@yahoo.co.jp) this is strange

2) the exposition of the problem, the main arguments, even the equations is complex and difficult to follow. An particularly nasty example is equation 5… where they explain the symbol xi(t) ..that does not appear directly into the equation!

3) The overall exposition is very non-direct. The first page of the introduction does not give any clue of what the authors are planning to do with the Flock problem.

In the discussion section the first paragraph is misleading (In this study, we discussed various critical phenomena of collective behavior produced by ambiguous interactions by constructing a model of ambiguous interactions based on the natural extension of the representative flocking model: quasi-attraction and quasi-alignment defined as short- and long-term predictions, respectively.” .. In fact they discuss one/two parametrs of a very restricted model of a very restricted problema.

Also in the Conclusion (The flock has dual aspects: a force distribution as fluctuation and a force converter as the group morphology (i.e., heterogeneous velocity distribution)….. the redaction is awkard it should say “Flock MODELS have two aspects ….”)

In general the text will benefit from a CONCEPTUAL editor. The authors do themselves weak favor with their text. For example this sentence “Although the OR and AND functions guide the understanding on PID” people could differ. PID is vastly different from simple logic gates.

Reviewer #3: The authors contribute important work and insights in the area of collective animal (and in general 'agent') behaviour.

They offer an extension of the standard 'boid' algorithm that addresses criticality as one of the most important but usually overlooked dynamical regimes necessary for sustainable collective behaviour. Therefore, this very well thought and worked out, paper deserves publication.

On this issue of criticality, I would like to point out to the authors another, maybe not so well known, publication that has addressed a certain extension of the standard Vicsek model for swarming based on a logistic map type of updating. What has been observed at that publication is the fact that sustainable, stable, swarming appears when the logistic-map updating self-organizes on the Feigenbaum critical point. Here is the reference:

A. García Cantú Ros, Ch.G. Antonopoulos, V. Basios,

Emergence of coherent motion in aggregates of motile coupled maps,

Chaos, Solitons & Fractals, Volume 44, Issue 8, 2011, Pages 574-586, ISSN 0960-0779,

https://doi.org/10.1016/j.chaos.2011.05.005. (https://www.sciencedirect.com/science/article/pii/S0960077911000683)

The above reference might help the authors find a relationship between critical phenomena and group

dynamics related to their important quest and as they quote on <<some questions="">association between individual criticality (i.e. Lévy-walk) and group criticality (i.e. scale-free correlation),

and how these group criticalities relate to collective turns.>>

I think that it will add to their discussion if they consider a connection with the above-mentioned paper.

But this is only out of collegial academic interest since, I would strongly suggest their work to be published

even if they choose not to discuss briefly or even mention this reference.

Their study is also interestingly original as they bring forth detailed relations between

scale-free correlations and collective rotations, this something new as far as I know.

They also provide the pseudocode for their simulations, as the journal's policy requires.

Personally, I would be happier if a free and open code was provided, lie an octave or R- code.

It is up to the authors to add this.

Another connection would be to discuss the authors' important work and take on the implications

that it has for inference as it is put forth here:

Gunji Yukio-Pegio, Murakami Hisashi, Tomaru Takenori and Basios Vasileios, 2018

Inverse Bayesian inference in swarming behaviour of soldier crabs

Phil. Trans. R. Soc. A.3762017037020170370

But again this is not to be taken as a requirement for adding any new reference, but rather for future discussion and work.

The paper is indeed opening new ground for such investigations, and therefore it merits publication even as is.</some>

**Have the authors made all data and (if applicable) computational code underlying the findings in their manuscript fully available?**

Reviewer #1: **No: **There is no available code to see how the simulaitons were run. There is attached a table of data, but these are just the processed numbers used in the final statistical analysis, but data from each run of the simulation is not available.

Reviewer #2: Yes

Reviewer #3: Yes

PLOS authors have the option to publish the peer review history of their article (what does this mean?). If published, this will include your full peer review and any attached files.

Reviewer #1: No

Reviewer #2: No

Reviewer #3: **Yes: **Vasileios Basios
---

## [Decision Letter · Decision Letter 1]

16 Dec 2022

Dear Dr. Niizato,

Thank you very much for submitting your manuscript "Functional duality in group criticality via ambiguous interactions" for consideration at PLOS Computational Biology. As with all papers reviewed by the journal, your manuscript was reviewed by members of the editorial board and by several independent reviewers. The reviewers appreciated the attention to an important topic. Based on the reviews, we are likely to accept this manuscript for publication, providing that you modify the manuscript according to the review recommendations.

Please carefully proofread your English.

Sincerely,

Xingru Chen, Ph.D.

Guest Editor

PLOS Computational Biology

Natalia Komarova

Section Editor

PLOS Computational Biology

Please carefully proofread your English.

Reviewer's Responses to Questions

**Comments to the Authors:**

Reviewer #2: THis second version is, without a doubt, an improvement with respect the first first version. But annoying mistakes remain in the use of the english language. For example in the first sentence of the DISCUSSION sentence ..

In this study, we showed the nested criticality that emerged from the same algorithm,

namely, the ambiguous interaction model, which is a natural extension of the

representative flocking model. We re-interpreted the attraction and alignment as short-

and long-term predictions, respectively, and as a key concept of our model, we

considered these predictions as regions C rather than points.

The expression "same algorithm" makes the sentence so difficult to understand.

also the expression "region C rather than points" ... could be written as "regions rather than points".

The authors need a conceptual editor ........not a mere "native speaker". They need someone with experience in technical english ....

Reviewer #3: The authors replied reasonably to all arguments and defend their thesis soundly, it deserves publication as its.

Thanks for providing the numerics in a type of source code and updating the text in almost all the issues raised before.

**Have the authors made all data and (if applicable) computational code underlying the findings in their manuscript fully available?**

Reviewer #2: Yes

Reviewer #3: Yes

PLOS authors have the option to publish the peer review history of their article (what does this mean?). If published, this will include your full peer review and any attached files.

Reviewer #2: **Yes: **Juan-Carlos Letelier

Reviewer #3: **Yes: **Vasileios Basios

Figure Files:

Data Requirements:

Reproducibility:

References:

---

## [Editor Report · Decision Letter 2]

10 Jan 2023

Dear Dr. Niizato,

We are pleased to inform you that your manuscript 'Functional duality in group criticality via ambiguous interactions' has been provisionally accepted for publication in PLOS Computational Biology.

Best regards,

Xingru Chen, Ph.D.

Guest Editor

PLOS Computational Biology

Natalia Komarova

Section Editor

PLOS Computational Biology

---

## [Editor Report · Acceptance letter]

1 Feb 2023

PCOMPBIOL-D-22-00874R2 

Functional duality in group criticality via ambiguous interactions

Dear Dr Niizato,

I am pleased to inform you that your manuscript has been formally accepted for publication in PLOS Computational Biology. Your manuscript is now with our production department and you will be notified of the publication date in due course.

With kind regards,

Anita Estes
